# The Usefulness of Calcium/Magnesium Ratio in the Risk Stratification of Early Onset of Renal Replacement Therapy

**DOI:** 10.3390/diagnostics12102470

**Published:** 2022-10-12

**Authors:** Rita Afonso, Roberto Calças Marques, Henrique Borges, Ana Cabrita, Ana Paula Silva

**Affiliations:** 1Nephrology Department, Centro Hospitalar Universitário do Algarve, 8000-836 Faro, Portugal; 2Department of Biomedical Sciences and Medicine, Universidade do Algarve, 8005-139 Faro, Portugal

**Keywords:** chronic kidney disease, calcium, magnesium, renal replacement therapy

## Abstract

Background: A growing number of studies have reported a close relationship between high serum calcium (Ca)/low serum magnesium (Mg) and vascular calcification. Endothelial dysfunction and vascular inflammation seem plausible risk factors for the enhanced progression of kidney disease. The aim of this study was to evaluate the role of the Ca/Mg ratio as a predictor of the early onset of renal replacement therapy (RRT). Methods: This was a prospective study conducted in an outpatient low-clearance nephrology clinic, enrolling 693 patients with stages 4–5 of CKD. Patients were divided into two groups according to the start of renal replacement therapy (RRT). Results: The kidney’s survival at 120 months was 60% for a Ca–Mg ratio < 6 and 40% for a Ca–Mg ratio ≥ 6 (*p* = 0.000). Patients who started RRT had lower levels of Hb, Ca, Mg, albumin, and cholesterol and higher values of phosphorus, the Ca/Mg ratio, and PTH. High values of phosphorus and the Ca/Mg ratio and low levels of Mg and GFR were independent predictors of entry into RRT. A high Ca/Mg ratio, high phosphorus levels, and low levels of GFR were associated with a cumulative risk for initiation of RRT. Conclusions: In our population, the Ca/Mg ratio is an independent predictive factor for the initiation of a depurative technique.

## 1. Introduction

Chronic kidney disease (CKD) is a worldwide public health problem, with an estimated global prevalence of 9.1%. The prevalence of CKD has increased by 29.3% in the last decades, with 2.5 million people receiving renal replacement therapy (RRT), a number that has been projected to double to 5.4 million in the next 10 years [1].

Traditional risk factors involved in the initiation and progression of CKD are established; however, the rate of progression shows considerable interindividual variability [2]. These findings suggest that other pathophysiological pathways and biological factors might be involved in the course of the disease [3]. The identification of new predictive markers might improve our understanding of the pathogenesis and prediction of which patients face an increased risk of CKD progression, providing the opportunity to implement preventive measures.

Prospective observational studies have identified new possibly predictive risk factors and markers associated with CKD progression, with growing evidence focusing on CKD–mineral and bone disorder (CKD–MBD) [4,5,6,7,8,9,10]. Despite the well-accepted abnormalities of bone and mineral metabolism in CKD, and the increased morbidity and mortality in haemodialysis patients [11,12,13,14,15], their impact on the renal outcomes in the pre-dialysis population is less well-characterized. Hyperphosphatemia, vitamin D deficiency, hyperparathyroidism, and high serum fibroblast growth factor 23 (FGF23) levels were found to predict CKD progression [8,16,17,18,19]. However, the role of serum magnesium (Mg) and calcium (Ca) levels in kidney disease progression is less clear.

Calcium is quantitatively the most abundant mineral in the body. Besides the structural role in the skeleton, calcium is a vital electrolyte that is required for many critical biological functions, including muscle contraction, vascular tone, nerve transmission, blood coagulation, and many enzyme-mediated processes [20]. Serum calcium levels in the higher range of normal have consistently been associated with greater vascular calcification and higher relative risks of death from all causes and from cardiovascular disease; however, no consistent or conclusive evidence currently prevails in the CKD population [21].

On the other hand, magnesium acts as a calcium blocker in vascular smooth cells. In addition to the protective effect of magnesium against pro-atherosclerotic risk factors, magnesium is a known vascular calcification inhibitor [22]. Over the past few decades, a number of experimental and observational studies have advocated for the role of magnesium on cardiovascular protection, in both the general population and CKD patients [23,24,25]. Given that cardiovascular and renal disease share similar etiopathogenic risk factors, it is plausible that the deficiency of magnesium also contributes to glomerular filtration rate (GFR) decline [26,27,28]. Recent studies indicate that hypomagnesemia may accelerate the deterioration of kidney function [28,29,30]. 

On the basis of this competitive interaction between calcium and magnesium, its balance may modify the kidney’s long-term outcomes. In a large Japanese incident haemodialysis cohort study, Hiroyuki et al. demonstrated that a high calcium–magnesium ratio was a predictor of all-cause and cardiovascular mortality [31]. However, to the best of our best knowledge, no study so far has assessed the association between calcium–magnesium ratio and the kidney’s survival in the pre-dialysis population. 

Therefore, in the present study, we evaluated, for the first time, the effect of calcium–magnesium ratio as a predictor of entry into renal replacement therapy (RRT) in a cohort of patients with CKD. 

## 2. Materials and Methods

An observational, prospective study involving 693 patients was conducted in the outpatient low-clearance nephrology clinic of the Centro Hospitalar Universitário do Algarve in Faro, Portugal, from 2010 to 2020. The primary endpoint of this study was start of renal replacement therapy (RRT). 

The study was submitted to and approved by the administration and ethics committee of the hospital. The study was conducted according to the principles of the Declaration of Helsinki, and study procedures were only performed after patients signed the informed consent. 

### 2.1. Patients

Adult patients with a diagnosis of stages 4–5 of CKD (>10 mL/min/1.73 m^2^ GFR < 30 mL/min/1.73 m^2^) were eligible to participate in this study. The CKD-EPI (Chronic Kidney Disease Epidemiology Collaboration) equation was used to estimate the glomerular filtrate rate (GFR). The exclusion criteria were: age < 18 years, GFR < 10 mL/min/1.73 m^2^ or ≥30 mL/min/1.73 m^2^, follow-up period of ≤3 months, uncontrolled hypertension (BP ≥ 140/90 mmHg), known neoplastic or infectious diseases, and acute cardiovascular events in last 6 months—defined as a history of one or more of the following: non-fatal myocardial infarction, angina pectoris (stable or unstable), and stroke or transient ischemic attacks or congestive heart failure.

### 2.2. Follow-Up

Patients returned on a regular basis for in-person nephrology consultation visits every three months. No patient was lost to follow-up within the observation period. Demographic, clinical, laboratory results, and medication data were collected from the clinical records.

### 2.3. Laboratory Measurements

Fasting blood samples were drawn from all subjects at the beginning of each consultation, and plasma/serum was frozen and stored at −80 °C. Biochemical variables, including serum calcium (Ca), phosphorus (P), magnesium (Mg), parathormon (PTH), albumin, creatinine, haemoglobin (Hg), and total cholesterol were measured. Serum Ca–Mg ratio was calculated by dividing the corrected Ca (mg/dL) by Mg (mg/dL). If the serum albumin level was <4.0 g/dL, the serum calcium concentration was corrected by the serum albumin level as follows: corrected Ca (mg/dL) = measured Ca (mg/dL) + 0.8 × [4 − albumin (g/dL)].

### 2.4. Statistical Analyses

Descriptive results were presented using mean and standard deviation (±SD) for continuous variables with normal distribution, using the Kolmogorov–Smirnov test. Patients were divided into two groups, according to the start of renal replacement therapy (RRT): G1, who did not start RRT, and G2, who had started RTT. Two-sample t-tests were used to assess differences in subgroups for continuous variables and chi-square tests for categorical variables. The Kaplan–Meier method for measuring kidney survival rate was applied, and a comparison between the two groups was based on the log-rank test. Univariate Cox regression analysis was used to identify independent factors associated with early onset of RRT. Statistically significant variables were analysed in multivariate Cox regression models to assess the main predictive risk factors for RRT initiation. Potential confounding factors offered to the logistic regression models included total cholesterol and albumin. The exponentials of the model parameters were the hazard ratio (HR) to other variables of the model, with 95% confidence interval. A modified Poisson regression with robust error variance estimation to calculate adjusted prevalence ratios (aPR) was used to estimate the cumulative relative risk of RTT initiation. Variables included in this multivariate analysis were age, gender, diabetes, Hg, P, Ca–Mg ratio, PTH, albumin, and GFR. The null hypothesis was rejected below the level of 5%. Differences were considered statistically significant for *p*-values < 0.05. Statistical software SPSS (version 17.0, Chicago, IL, USA) for Windows was used for statistical data analysis. 

## 3. Results

A total of 693 patients meeting the inclusion criteria with stages 4–5 of CKD were evaluated during a 120-month period from January 2010 to December 2020. The mean age was 70.09 ± 12.51 years (range: 25–96), and 53.5% were female. The mean GFR was 19.91 ± 8.11 mL/min/1.73 m^2^. A total of 62% and 30% of patients had hypertension and diabetes, respectively. Table 1 describes the patients’ mean clinical and biochemical characteristics, including osteo-mineral markers. 

The patients were allocated into two groups according to the start of RRT. Group 1, the non-RRT group, encompassed 541 patients, and group 2, the RRT group, included 152 patients. All the assessed variables presented significant differences between the two groups. As presented in Table 2, the RRT group displayed significantly lower serum levels of Hb (*p* = 0.000), Ca (*p* = 0.000), Mg (*p* = 0.000), albumin (*p* = 0.039), and total cholesterol (*p* = 0.019), as well as significantly higher values for phosphorus (*p* = 0.000), the Ca–Mg ratio (*p* = 0.000), and PTH (*p* = 0.000).

The association between the levels of the calcium/magnesium ratio and the start of renal replacement therapy using the chi-squared test showed that higher levels of the calcium/magnesium ratio were associated with the start of a depurative technique. The median follow-up between who started RRT and who did not start RRT were 27.9 ± 19.7 and 31.59 ± 19.3 months (*p* = 0.04), respectively.

Variables associated with RRT initiation in the univariate Cox regression analysis (Table 3) were used in the multivariate Cox regression model (Table 4). The results clearly show that higher levels of phosphorus and Ca/Mg ratio (HRa = 1.638, *p* = 0.001; HRa = 1.292; *p* = 0.002, respectively) and lower levels of Mg and GFR (HRa = 0.761, *p* = 0.005; HRa = 0.934, *p* = 0.0001, respectively) correlated with kidney survival and RRT initiation.

Furthermore, the Poisson regression analysis showed that high values for the Ca–Mg ratio and phosphorus are cumulative risk factors for entry into RRT (aPR = 1.986; 95% CI 1.026–3.051; *p* = 0.002; aPR = 1.607; 95% CI 1.324–1.950; *p* < 0.0001, respectively). Additionally, low levels of GFR (aPR = 0.927; 95% CI 0.891–0.964; *p* < 0.0001) were associated with initiation of RRT (Table 5).

ROC curve analysis demonstrated that a Ca–Mg ratio > 6 is the best cut-off value associated with RRT initiation (AUC = 0.662; 95% CI 0.613–0.711). Using Kaplan–Meier analysis (Figure 1), it was observed that the kidney’s survival at 1 year was 60% for a Ca–Mg ratio < 6 and 40% for a Ca–Mg ratio ≥ 6. Additional risk for RRT initiation decreases with a lower Ca–Mg ratio. The log-rank test confirmed the existence of significant differences between the two groups (*p* = 0.000).

## 4. Discussion

Mineral and bone disorder (MBD) is one of the important complications caused by chronic kidney disease and is associated with mortality in patients undergoing maintenance dialysis [11,12,13,14,15]. However, the impact of abnormal bone and mineral metabolism in kidney disease progression, for pre-dialysis patients, is still not fully elucidated. In this large-sample study with 693 stages 4–5 CKD patients, we found that higher values for serum phosphorus and the calcium/magnesium ratio and lower levels of serum magnesium were strongly and independently associated with entry into renal replacement therapy. Serum calcium was not associated with a depurative technique event. These results suggest the potential importance of the calcium/magnesium ratio as a new marker associated with an early start of RRT, enabling early identification and interventions of patients most likely to progress to end-stage renal disease (ESRD). 

The direct association between plasma phosphate concentration and a decline in both renal function and renal morphological changes has been extensively reported and can be explained pathophysiologically by the “precipitation-calcification hypothesis” [17,32,33,34,35,36,37,38,39]. Hyperphosphatemia results from failure in the excretion of the dietary phosphate load by the diseased kidney [40], leading to persistent hyperparathyroidism, decrease in serum ionized calcium, mobilization of calcium from bone, and, consequently, an elevated serum [Ca] × [P] product. Supersaturation of [Ca] × [P] product, which is characteristic of the uremic state, favours metastatic calcification [41]. In CKD animal models, a high plasma phosphate concentration leads to the deposition of calcium phosphate crystals in the mitochondria of tubular cells and renal interstitium. This causes oxidative stress, inflammation, cell tubular injury, and mitogenesis of fibroblasts, resulting in progressive loss of renal function [33]. Gimenez et al. demonstrated that renal calcium deposition in human renal biopsies was correlated with hyperphosphatemia. Further evidence is given by the use of a low-phosphate diet or the use of non-calcium-containing phosphate binders in rat models of CKD, showing a reduction in intrarenal calcium phosphate deposition and interstitial fibrosis, which improves renal histology and attenuates the deterioration of kidney function [42,43,44]. This suggests that phosphate has an etiological role per se in intrarenal calcification, contributing to nephrocalcinosis. Furthermore, vascular calcification is an active cell-mediated process, and phosphorus has been shown to be an important mediator in the development of renal artery calcification, which is another mechanism whereby higher serum phosphorus levels may contribute to progressive loss of renal function [45].

Conversely, Mg has an essential protective role in the calcification milieu. As a natural Ca channel antagonist, Mg has both a passive inorganic phosphate (Pi)-buffering role, preventing calciprotein particles (CPP) maturation and hydroaxyapatite formation, as well as an active role, preventing osteogenic vascular smooth muscle cells (VSMCs) transdifferentiation [46,47]. As Mg has an anticalcification property, it is conceivable that Mg might exert beneficial effects on renal prognosis by alleviating calcium phosphate toxicity and, consequently, preventing CKD progression [48,49]. Thus, hypomagnesemic patients are at risk. 

On the basis of the competitive interaction between Ca and Mg, its balance may modify the kidney’s long-term outcomes. Indeed, to better understand the impact of Ca and Mg on vascular calcification and renal outcomes, both should be considered together rather than separately. In this field, serum Ca–Mg ratio has gained attention in recent years, since a ratio above 2:1 has been associated with increased risk of metabolic, inflammatory, and cardiovascular disorders [30]. In a large Japanese incident haemodialysis cohort study, Hiroyuki et al. showed that a high Ca–Mg ratio was a predictor of all-cause and cardiovascular mortality [31]. Interestingly, this study was the first to demonstrate the impact of the Ca–Mg ratio in the area of nephrology. Additionally, Park et al. demonstrated that a higher calcium–magnesium ratio in hair was associated with a greater coronary artery calcification. Moreover, and in agreement with this observation, the authors proposed the use of high calcium–magnesium ratio as a putative cardiovascular risk factor for the prevention and early diagnosis of cardiovascular disease [50]. In this line, *in order to improve prediction of which patients are at risk of CKD progression, we evaluated the role of the calcium–magnesium ratio in the kidney’s survival.*

Finally, beyond Mg being known as a vascular calcification inhibitor, it has also been associated with dyslipidemia, hypertension, insulin resistance, inflammation, oxidative stress, and sympathetic overactivity; all of which are correlated with an increased risk of cardiovascular events [26,27,28]. Tin and colleagues [28] demonstrated that low serum Mg levels were associated with incident CKD and ESRD in a cohort of 13,226 patients (GFR > 60 mL/min/1.73 m^2^), with a median follow-up period of 21 years. Compared with patients with serum Mg levels of ≥0.9 mmol/L, those with serum Mg levels of ≤0.7 mmol/L had a 1.58- and 2.39-fold higher risk of CKD and ESKD, respectively. These associations remained significant after excluding users of diuretics and across subgroups stratified by hypertension, diabetes, and self-reported race. To assess the impact of Mg on the progression of CKD, Van Laecke and colleagues conducted a retrospective cohort study of 1650 patients with CKD with a median follow-up time of 5.1 years. This study demonstrated that each 0.1 mg/dL increase in baseline serum Mg concentration was associated with a 7% decreased risk of death, after adjustment for potential cofounders such as diabetes and hypertension. Moreover, the GFR declined faster among patients with lower baseline serum magnesium levels [26]. However, the significance of this association was attenuated/disappeared after adjusting for diuretic use in particular. Since magnesium is closely involved in the pathogenesis of diabetes mellitus, Silva et al. examined 191 type 2 diabetic patients with mild–moderate CKD and showed that hypomagnesemia can be a novel predictor of ESRD in this population. A statistically significant decrease in the risk of progression to RRT with the high-magnesium group (Mg > 2.3 mg/dL) and an increased risk with the low-magnesium group (Mg < 1.2 mg/dL) were demonstrated [25]. These data reinforce that Mg deficiency can increase the risk of not only cardiovascular disease but also kidney disease. 

Overall, these findings suggest a tight and synergistic effect between high levels of P and Ca and low levels of Mg, in vascular calcification and renal outcomes. This study expands our knowledge of nontraditional risk factors involved in the early onset of renal replacement therapy. 

The strengths of our prospective study include the large sample size of CKD patients and long follow-up time. To the best of our knowledge, this is the first study to evaluate the role of the calcium–magnesium ratio in the kidney’s survival in pre-dialysis patients. Additionally, all serum calcium and magnesium values were evaluated during follow-up, rather than just the baseline values, and findings remained consistent. However, this study also has some limitations that should be considered. First, the data came from a single centre, and only patients with the most advanced stage of renal disease before RRT were included in this study. This may limit the generalizability of our results. Nevertheless, an advantage of investigating pre-dialysis patients is the large variation in both declines in renal function and biochemical abnormalities due to uraemia. These increase the likelihood of detecting risk factors, which may be too weak to be detected in patients with a less severe disease. Second, no data were collected related to dietary Mg intakes and urinary Mg excretion, so, consequently, it was not possible to correlate dietary intake with serum magnesium levels. Third, proton pump inhibitors and diuretic drugs were not included on the list of pharmacological medications analysed, which can interfere with calcium and magnesium homeostasis.

## 5. Conclusions

In conclusion, this study shows that low serum Mg levels and high phosphorus and Ca–Mg ratio values are independent and cumulative predictive factors for the start of a depurative technique in patients with stages 4–5 of CKD. The high diagnostic value found for the Ca–Mg ratio in CKD progression, together with the previous knowledge that the Ca–Mg ratio is associated with metabolic, inflammatory, and cardiovascular disorders, including vascular calcification, suggest that the Ca–Mg ratio is a novel marker with the potential utility to estimate the clinical risk of kidney disease progression. Additional molecular and clinical research is required to validate the use of this non-invasive measured marker in the field of nephrology.

## Figures and Tables

**Figure 1 diagnostics-12-02470-f001:**
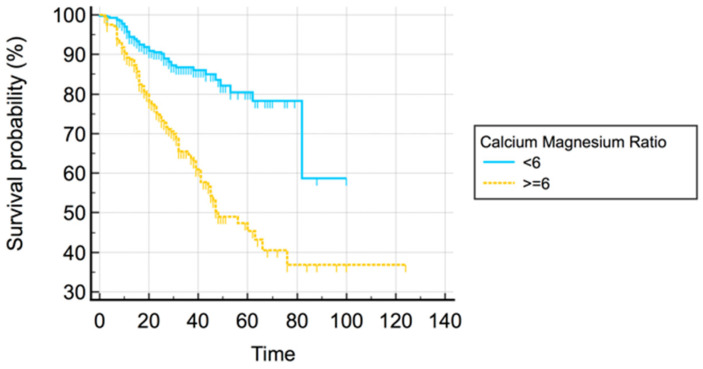
Kaplan–Meier survival analysis.

**Table 1 diagnostics-12-02470-t001:** Baseline patient characteristics.

General Characteristics	Values
Number of patients, *n*	693
Age (years)	70.09 ± 12.51
Gender f/m (%)	371/322 (53.5/46.5)
Hb (g/dL)	11.57 ± 1.12
Ca (mg/dL)	9.25 ± 0.67
Albumin (g/dL)	3.98 ± 0.47
P (mg/dL)	4.06 ± 0.76
Mg (mg/dL)	1.81 ± 0.71
Ca–Mg ratio	6.13 ± 2.76
PTH (pg/mL)	238.03 ± 185.37
Total cholesterol (mg/dL)	180.81 ± 45.05
GFR (mL/min)	19.91 ± 8.11
Cr (mg/dL)	3.13 ± 1.24
Hypertension (%)	62.3
Diabetes (%)	29.6
Coronary artery disease (%)	8.8
Cerebrovascular disease (%)	8.4
ACE inhibitor/ARB (%)	47.3/56
CCBs (%)	41.4
Darbepoetin alfa (%)	54.7
Vitamin D analogue (%)	20.5
Phosphate binders (non-calcium-based phosphate binders) (%)	12.8

Hb, haemoglobin; Ca, calcium; P, phosphorus; Mg, magnesium; Ca–Mg ratio, calcium–magnesium ratio; PTH, parathormon; GFR, glomerular filtration rate; Cr, creatinine; ACE inhibitor, angiotensin-converting-enzyme inhibitor; ARB, angiotensin-receptor blocker; CCBs, calcium channel blocker; P binders, phosphate binders.

**Table 2 diagnostics-12-02470-t002:** Demographic and laboratory characterization of the population during the 10 years of follow-up (mean values). Comparison of variables among the two groups.

Variable	G1: Start RRT (No) (*n* = 541/693)	G2: Start RRT (Yes) (*n* = 152/693)	*p* Value
Age (years)	70.85 ± 12.05	67.38 ± 13.73	0.005
Hb (g/dL)	11.75 ± 1.06	10.95 ± 1.09	0.000 *
Ca (mg/dL)	9.34 ± 0.52	8.95 ± 0.98	0.000 *
Albumin (g/dL)	4.00 ± 0.39	3.88 ± 0.66	0.039 *
P (mg/dL)	3.88 ± 0.56	4.69 ± 0.99	0.000 *
Mg (mg/dL)	1.92 ± 0.68	1.40 ± 0.67	0.000 *
Ca–Mg ratio	5.73 ± 2.62	7.56 ± 2.75	0.000 *
PTH (pg/mL)	209.71 ± 165.91	338.84 ± 214.33	0.000 *
GFR (mL/min)	21.37 ± 8.14	14.73 ± 5.49	0.000 *
Total cholesterol (mg/dL)	183.17 ± 42.86	172.39 ± 51.38	0.019 *

Continuous variables are expressed as a mean ± standard deviation. Hb, haemoglobin; Ca, calcium; P, phosphorus; Mg, magnesium; Ca–Mg ratio, calcium–magnesium ratio; PTH, parathormon; GFR, glomerular filtration rate. * Statistically significant (*p* < 0.05).

**Table 3 diagnostics-12-02470-t003:** Univariate Cox regression analysis results for RRT initiation.

Independent Variable	G2: Start RRT (Yes)		
	ß	HR (95% CI)	*p* Value
Age	0.018	1.083 (1.009–2.994)	0.004
Hb	−0.733	0.481 (0.412–0.561)	0.000
Ca	0.449	1.638 (1.276–3.707)	0.000
P	0.978	2.658 (2.267–3.117)	0.000
Mg	−0.946	0.388 (0.301–0.502)	0.000
Ca–Mg ratio	0.171	1.186 (1.127–1.249)	0.000
CaxPi	0.871	1.090 (0.990–1.111)	0.065
PTH	0.002	1.002 (1.002–1.003)	0.000
GFR	−0.158	0.854 (0.828–0.882)	0.000

HR, hazard ratio; CI, confidence interval; RRT, renal replacement therapy; Hb, haemoglobin; Ca, calcium; P, phosphorus; Mg, magnesium; Ca–Mg ratio, calcium–magnesium ratio; PTH, parathormon; GFR, glomerular filtration rate. Statistically significant (*p* < 0.05).

**Table 4 diagnostics-12-02470-t004:** Multivariate Cox regression analysis results for RRT initiation.

Independent Variable	G2: Start RRT (Yes)		
	ß	HR (95% CI)	*p* Value
Age	−0.012	0.988 (0.975–1.001)	0.075
Hb	−0.549	0.578 (0.489–2.682)	0.134
Ca	0.320	0.726 (0.531–1.004)	0.056
P	0.493	1.638 (1.240–2.164)	0.001
Mg	−0.059	0.761 (0.493–0.859)	0.005
Ca–Mg ratio	0.509	1.292 (1.033–2.981)	0.002
PTH	0.001	1.001 (0.905–1.001)	0.160
GFR	−0.069	0.934 (0.900–0.968)	0.000

HR, hazard ratio; CI, confidence interval; RRT, renal replacement therapy; Hb, haemoglobin; Ca, calcium; P, phosphorus; Mg, magnesium; Ca–Mg ratio, calcium–magnesium ratio; PTH, parathormon; GFR, glomerular filtration rate. Statistically significant (*p* < 0.05).

**Table 5 diagnostics-12-02470-t005:** Modified Poisson regression analysis.

Variable	G2: Start RRT (Yes)		
	ß	aPR (95% CI)	*p* Value
Age	−0.005	0.995 (0.985–1.006)	0.375
Hb	−0.212	0.809 (1.020–0.909)	0.127
Albumin	−0.063	0.939 (0.688–1.282)	0.693
P	0.474	1.607 (1.324–1.950)	0.000
Ca–Mg ratio	0.014	1.986 (1.026–3.051)	0.002
PTH	0.001	1.001 (0.800–1.001)	0.134
GFR	−0.076	0.927 (0.891–0.964)	0.000

aPR, adjusted prevalence ratio; CI, confidence interval; RRT, renal replacement therapy; Hb, haemoglobin; Ca, calcium; P, phosphorus; Mg, magnesium; Ca–Mg ratio, calcium–magnesium ratio; PTH, parathormon; GFR, glomerular filtration rate. Statistically significant (*p* < 0.05).

## Data Availability

Not applicable.

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
