# Peer review of "The Usefulness of Calcium/Magnesium Ratio in the Risk Stratification of Early Onset of Renal Replacement Therapy"

_diagnostics, 2022, doi:10.3390/diagnostics12102470_

Round 1

Reviewer 1 Report

Rita Serra Afonso and collages collected clinical data on 693 patients with stage 4-5 CKD and compeer two groups of patients from the pre-dialysis clinic according to  starting of renal replacement therapy studding the effect  of Ca-Mg ratiolevels , Hb, Ca, Mg, P, albumin and cholesterol as a predictors of entry into dialysis.

This data is over all interesting and well known in the nephrology literature that P/Ca are predictive factors to severity of renal failure ,lower GFR, higher PTH (just as in this study) but not a  risk factors. I don't think that this study show that this are risk factors and should be changed.  

Please give use more details on the use of phosphate binders (Ca based phosphate binders or non Ca based) and the use of vitamin D analogs as a cause for this differences .Low Mg can be secondary to the used of  diuretic drug that are needed in higher dose more before RRT so data on furosemide should be added. 

Minor revision: Please don't use ESRD the use of CKD stages terminology and RRT is better. 

The recommendation on the use of low Posphorus diet should be avoided, since its can mean a low protein diet  that cannot be recommended by this retrospective study. Please also add dietary assessment commonly used in pre-dialysis patients to assessed the effect of protein P, Ca ,Mg intake. 

Reviewer 2 Report

This is a 10-year retrospective hospital-based cohort from January 2010 to December 2020. The study includes a total of 693 patients with CKD stage 4-5. Patients were further divided into two groups according to the development of ESRD and start of renal replacement therapy. The relationship between Ca-Mg ratio and kidney disease progression in a cohort of patients with CKD, not yet on dialysis was examined. Some problems needed to be addressed more. My comments are listed below.

1.      Is it unclear how Ca-Mg ratio is determined? Was the first record of the Ca-Mg ratio used by the author? Using the first record of Ca-Mg ratio to predict future 10-year risks may have some drawbacks, since Ca-Mg ratios change over time. Performing time-varying Ca-Mg ratio may be the better methodology for this issue.

2.      Is there a reason why the Ca-Mg ratio is set at 6? An optimal cutoff value can be calculated based on the ROC curve

3.      The statements about group 1 and group 2 should be changed to ESRD or non-ESRD groups to avoid confusions.

4.      Because group 1 and group 2 had much different clinical characteristics. For matching, propensity scores for these two groups approach may be considered.

5.      Why dose this study did not have underlying comorbidities, such as diabetes mellitus, coronary artery disease, or hypertension?

6.      Further details about the amount (proportion) of missing data should be presented

7.      Over a 10-year period, only 152 patients with stage 4-5 CKD received RRT (21.9%). The proportion seemed a little low in advanced CKD. What is the median follow-up between those started RRT or did not start RRT?

8.      Furthermore, why did the authors include advanced CKD stages 4-5 rather than CKD stages 3 for follow-up? As CKD progresses to stage 3, calcium and PTH disturbances may gradually already appear.

9.      In several places, renal outcomes were mentioned in the manuscript. However, renal outcomes is defined differently, such as kidney disease progression, ESRD, and renal replacement therapy. In addition, I didn’t find the study to explore kidney disease progression (is the definition is based on renal function decline?) Which one is really used in this manuscript ? and the definition should be clarified in the method of manuscript.

10.  The author mentioned that “Hiroyuki et al. demonstrated that a high calcium:magnesium ratio was a predictor of all-cause and cardiovascular mortality in a large Japanese incident hemodialysis cohort study”. How about the risks of all-cause and cardiovascular mortality in this study (in such advanced CKD patients)?

11.  When calculating eGFR, this study used CKD-EPI formula. Recently, the newest, race-free CKD-EPI equations (Inker, NEJM 2021) was used. The race-free CKD-EPI equations may be used in this manuscript.

12.  How does the multivariate analysis handle? How and which variables were included in the multivariate analysis?

13.  Table 4 shoud added Ca x IP ratio in the multivariabe Cox analyses to compare the effects of Ca-Mg ratio.
